# Refund of Consumption Tax to Low-Income People: Impact Assessment Using Difference-in-Differences

Jorge Luis Tonetto [1,*] , Adelar Fochezatto [1] and Giovanni Padilha da Silva [2]

1   Business School, Pontifícia Universidade Católica do Rio Grande do Sul, Porto Alegre 90610-970, Brazil; adelar@pucrs.br
2   Rio Grande do Sul State Revenue Office, Porto Alegre 90030-080, Brazil; giovannis@sefaz.rs.gov.br
*   Correspondence: jorge.tonetto@pucrs.br; Tel.: +34-675-209516

**Abstract:** One way to reduce inequality and poverty is to promote tax justice. In 2021, the government of the state of Rio Grande do Sul, Brazil, implemented a program (the Devolve-ICMS Program) that refunds consumption tax to low-income citizens (cashback). This study aims to evaluate the impacts of this Program using a differences-in-differences model and having, as response variables, the monthly sum of electronic invoices issued to the Program's beneficiaries, as well as their value. The database used includes all invoices issued to the target population during the 12 months before the Program's implementation and the 14 months after its implementation, resulting in 7.7 million records. To receive the tax refund, the eligible population must pick up a Citizen Card, made available by the state government, which was done by a significant part of this population. The treatment group is composed of eligible citizens who have the Card, whereas the control group comprises eligible citizens who do not have it. The results show that the Program is effective, as it has reduced tax pressure on poor people and increased both the number of invoices issued and their value.

**Keywords:** poverty; inequality; tax justice; difference-in-differences



## 1. Introduction

Brazil is a very unequal country, with high rates of poverty and labor informality. Using the World Bank's concept of the social poverty line, Bagolin et al. (2022) point out that the social poverty rate in Brazil reached 30.4% in 2021. Thus, 64.6 million Brazilians can be considered socially poor. Regarding labor informality, IBGE (2022) reports that there are discrepancies between the country's regions. In 2021, informal employment levels were predominant in the north (58.6%) and northeast (55.9%) regions, while lower levels were verified in the southeast (33.9%) and south (26.8%) regions.

This informality constitutes an important source of inequalities, as these people have no access to social protection mechanisms, such as the right to retirement and paid maternity or sick leave.

The implementation of social programs such as Bolsa Família managed to improve this situation for some time, but it has deteriorated in recent years. Dependence on benefits from social programs nearly doubled from 2012 to 2020. In this period, there was an increase in the income vulnerability of the Brazilian population, with an increased proportion of people in the lower income brackets. Considering the per capita household income, the population in the 10% poorest bracket was the one that lost the most in this period (IBGE 2022).

Considering that, according to the World Bank, Brazil is an upper middle-income country, but with profound income inequality and high rates of poverty and labor informality, it is imperative to advance in the implementation of public policies that contemplate social inclusion and shared economic growth. However, given the high regional heterogeneity, it is possible that more focused programs—in terms of target populations and the regions where they live—be more effective than universal programs. The challenge of combating

inequality often rules out miraculous solutions. As discussed by Banerjee and Duflo (2021), there is a need to understand where hope lies and why symbolic subsidies can have more than symbolic effects.

The Devolve-ICMS Program, the object of this study, is inserted in this context. It was implemented by the government of the state of Rio Grande do Sul in 2021 and consists of directly returning to the low-income population part of the amount corresponding to the tax levied on their purchases, based on the invoices issued in their Taxpayer Registration Number (CPF). It is, therefore, a tax customization that aims to mitigate tax regression and improve the living conditions of citizens, in addition to encouraging their engagement in a process of greater exercise of fiscal citizenship. The people targeted by this Program include those on the Bolsa Família Program register, as well as those selected according to other criteria (see Figure 1). There are similar experiences in other countries, mainly in Japan and Canada; however, to the best of our knowledge, this is the first tax personalization application to use the electronic invoice to quantify the consumption of beneficiary families.

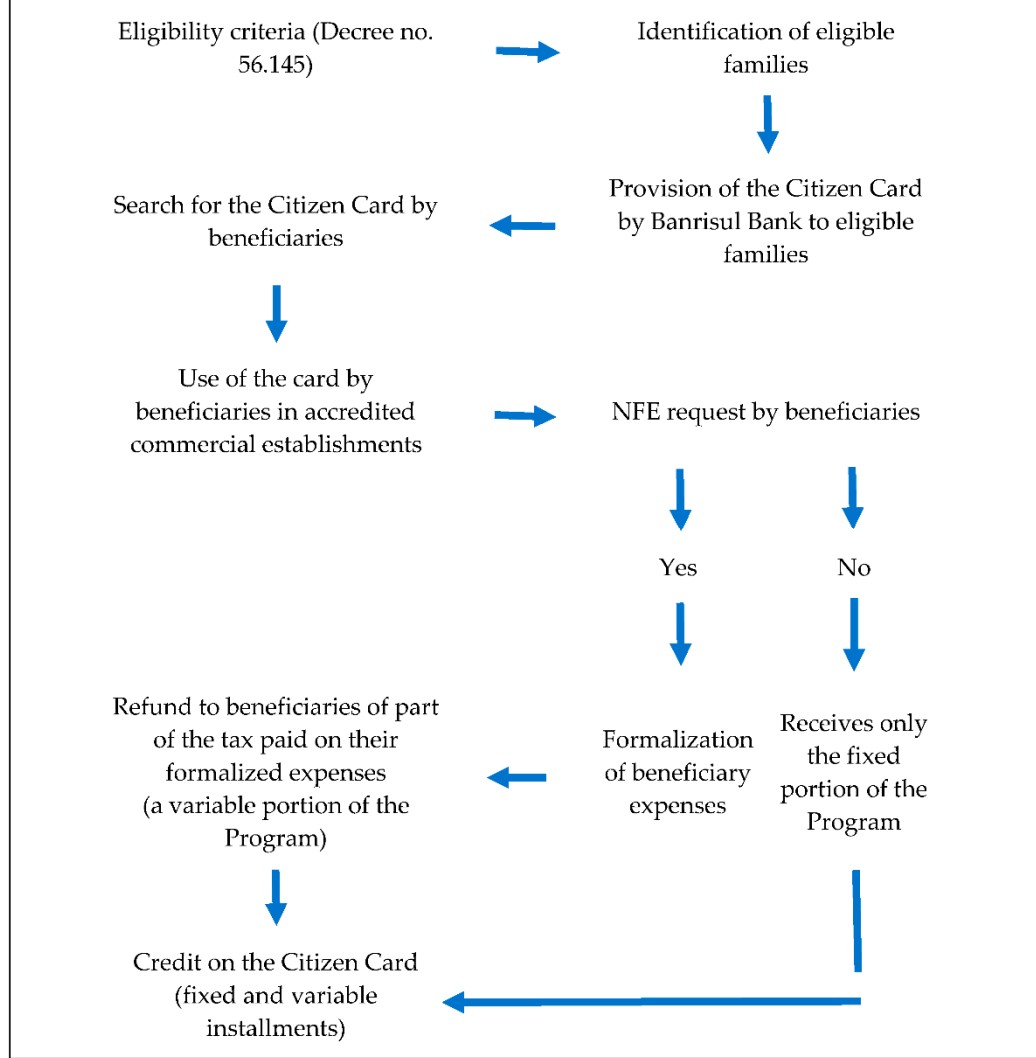

**Figure 1.** Schematic illustration of the functioning and operationalization of the Devolve-ICMS Program. Source: Prepared by the authors.

This study aims to evaluate the impacts of the Devolve-ICMS Program on the consumption value of the beneficiaries and their participation in the tax issuance program. In addition, an estimate is made of the change in tax pressure by income bracket after the implementation of the Program.

To this end, the difference-in-differences method was applied. Program beneficiaries, in addition to meeting the eligibility criteria, need to pick up a card (Citizen Card), made available by Banrisul (the bank of the state of Rio Grande do Sul). Eligible people who did not pick up the card comprised the control group. The analyzed periods were the 12 months before and the 14 months after the implementation of the Program.

The results indicate that the Program has reduced the tax pressure on the low-income population. In addition, it increased the consumption of this segment of the population and increased the number of invoices issued in their name. These results show that the Program is effective in improving the living conditions of the most vulnerable population in the state of Rio Grande do Sul and, consequently, reducing inequalities. This is the first impact assessment of the Devolve-ICMS Program using the difference-in-differences method.

In addition to this Introduction, this manuscript is composed of Section 2, which describes the Devolve-ICMS Program in detail. Section 3 presents a bibliographic review of the use of fiscal policies to face poverty and inequality, as well as other experiences in similar programs. Section 4 describes the methodology, presenting details on the database and the econometric models used. In Section 5, the results found are analyzed. Finally, Section 6 presents the main conclusions of this study.

## 2. The Devolve-ICMS Program

One of the characteristics associated with indirect taxes on consumption, such as the Tax on the Circulation of Goods and Services (ICMS), is their regressive nature; that is, the burden is proportionally greater on the income of relatively poorer families. Aiming to face this dysfunction, as of the 1980s, the technique of "differentiation" between tax burdens began to be widely used, with the objective of recording, in a more lenient way, the items with greater participation in the consumption structure of the poorest families. A classic example of this policy is the exemption of "basic food baskets". However, this technique soon began to be questioned because of its ineffectiveness in mitigating progressiveness. Given this, the government of the state of Rio Grande do Sul created the Devolve-ICMS Program, which presents a more effective logic to face the inequity of this tax, and favor income redistribution.

The tax administration office of the state of Rio Grande do Sul has evolved in the incorporation of innovations to its taxation and collection systems. The creation, 10 years ago, of the Electronic Invoice (NFE), which, having gone through several phases, today reaches the final consumer, is considered a milestone. Currently, 3 million people are registered in the Rio Grande do Sul invoice program, which represents more than 25% of the state's population. An important stimulus to citizen participation is its system of prizes and monetary incentives. Every time a consumer requests the inclusion of their CPF in the tax document, points that are valid for monthly and instant raffles are generated. The expansion of the digital world through the Internet and data exchange has provided tax authorities with an enormous amount of information. New tools with immediate interaction have brought the tax authorities closer to the taxpayers and increased citizen participation, helping to combat tax evasion and unfair competition.

In this context, the Devolve-ICMS Program was implemented by the government of the state of Rio Grande do Sul, Brazil, in November 2021. It is an innovative program that has, as references, other similar programs and the doctoral thesis by da Silva (2017)—an employee of the State Revenue Office—that analyzed the personalization of the consumption tax. The personalization of this type of tax is still not widespread, but it has progressively gained visibility in recent years, both in the academic and political fields. In Brazil, this topic has been included on the agenda of the tax reform currently under discussion in the National Congress. In the Brazilian context, therefore, the Devolve-ICMS Program represents the first experience of tax customization.

The Devolve-ICMS Program consists of directly refunding the amounts corresponding to the tax levied on purchases formalized in NFEs to the low-income population. Thus, the Program directly returns to low-income people part of the amount corresponding to the

tax incurred on their purchases, based on the invoices issued in their name (CPF). To the best of our knowledge, this is the first tax personalization application to use the NFE to quantify the consumption of beneficiary families and serve as a basis for tax refunds.

The target population of the Devolve-ICMS Program consists of families with income of up to three national minimum wages, or monthly per capita income smaller than half the national minimum wage. According to Decree no. 56.145 of 20 October 2021, these are the following requirements to participate in the Program: I—declared monthly per capita family income smaller than half the national minimum wage, or declared monthly family income of up to three national minimum wages; II—domiciled in the state of Rio Grande do Sul; III—head of household with an active Individual Taxpayer Registration Number (CPF); and IV—family unit that fits into at least one of the following hypotheses: (a) be a beneficiary of the *Bolsa Família* Program, provided for in Federal Law no. 10.836 of 9 January 2004; (b) having a family member enrolled in regular secondary education in a state public school. Therefore, the Program aims to make state taxation more modern and fairer by reducing the tax burden on the neediest families.

Tax refunds occur through a bank card provided by Banrisul (the bank of the state of Rio Grande do Sul), called Citizen Card. This possibility of return was approved in 2020 by the Rio Grande do Sul House of Representatives through Law No. 15.576 of the Tax Reform. Cards are still being delivered, as there are still a significant number of beneficiaries who have not picked them up yet. In addition, updates to the Cadastro Único[1] require the printing and delivery of new cards. The card can be used in more than 140,000 establishments throughout the state, such as supermarkets, bakeries, pharmacies, and others, also strengthening the local economy. Figure 1 illustrates the functioning and operationalization of the Devolve-ICMS Program.

The Program was designed to be implemented in stages. The first stage began in September 2021 and the second in July 2022. In the first stage, 432,000 families benefited, with a return of fixed installments of BRL 100.00 per quarter. In the second stage, the number of families covered increased to 527,000, and, in addition to the fixed portion, they began to receive a variable portion determined by the amount of expenses formalized in the NFEs.

The monetary incentives of the Program are shaping new taxpayer habits. To support behavior understanding, Fehr et al. (2015) present a behavior matrix (Behavior Change Matrix) based on empirical research that shows that contributions to the public good depend on two conditions: awareness of a social norm to contribute and willingness to contribute. To induce or educate people's behavior, several measures can be used, such as monetary incentives, educational measures, and "nudges". These measures can be very effective, but they depend on the specific contexts of the target population.

## 3. Theoretical and Empirical Review

Regarding the use of public policies based on consumption taxation, it is worth highlighting the solutions adopted in Japan and Canada, which are similar to the Devolve-ICMS Program implemented in Rio Grande do Sul. Despite the similarity regarding the tax refund to poorer people and families, it is important to highlight the profound differences between these countries regarding the importance of value-added tax (VAT). These two countries are among those that collect the least VAT as a percentage of their GDP and practice relatively low rates. Currently, Canada has a collection of 4.7% of the GDP, representing 13.6% of the amount collected, while Japan collects 4.9% of the GDP, corresponding to 14.9% of its collection (OECD 2022).

Japan has implemented a social inclusion program via VAT. For administrative convenience, the "customization" of the tax aims to produce exemptions in the consumption of specific products, corresponding to combinations between product and consumer, to eliminate the revenue losses typical of the general application of products to all consumers. This solution establishes variations in VAT, according to different combinations of products and

consumers. This increases the complexity for companies to manage their tax obligations (Barreix et al. 2012).

Canada has opted for a simpler solution, which applies a uniform rate with exemptions for basic items, such as food and health-related products. To reinforce the tax reduction for the most vulnerable populations, it included a partial compensation mechanism for the tax passed on to the consumption of these groups. The amount of the transfer is defined based on marital status, number of family members, and taxpayer income level. It is simpler than the Japanese solution, as it is carried out in a single compensatory transfer operation, and not on each consumption operation (Barreix et al. 2012). In Canada, the administration of benefits, which are in the form of credits, is the responsibility of the Canada Revenue Agency, which interacts in partnership with the federal, provincial, and territorial spheres. The credit is granted quarterly to low-income individuals and families to offset all, or part of, the tax paid on goods and services or harmonized sales tax. Eligibility is determined using information from the Personal Income Tax and Statement of Benefits.

Another alternative is the creation of a digital VAT (D-VAT) with the aim of combating regressivity, using technology to individualize consumption. This solution includes invoicing, declaration, and collection in real-time. There is a biometric identification of consumers, which allows the holder to be exempt from tax at the time of transaction, with minimal risk of fraud. Their purchases are taxed at a zero rate and do not accumulate VAT from previous phases. This proposal requires a massive investment in information systems, which would only be available to developed countries. The use of a D-VAT card may present problems in terms of user privacy (Barreix et al. 2012).

A great advantage of personalized VAT is that it tends to reduce tax regressivity, which ends up cooling the efficiency versus equity debate (da Silva 2017). However, the viability of a proposal will depend on the country's, or subnational's, poverty conditions, as well as on the funding sources and technical implementation conditions. There may be cases where the tax base is so eroded that a simple 1% increase in the rate will already allow the generation of resources to improve the situation of poverty and regression. However, if the poverty level is indeed very high, input from other fund sources may be necessary. Another possible source may be the inefficiently applied exemptions, whose reduction can generate enough resources to apply personalized VAT, gaining efficiency and equity.

When implementing public policies, one of the relevant errors refers to the issue of focusing. Barreix et al. (2012) point out that the inclusion error can extend benefits to groups that have the highest income in society, and this inclusion error configures a focusing error. The tax offset must consider the amount of the tax reduction or refund and the individuals who will benefit from it.

The objectives of the program must be conceptually defined: whether it will only seek to reimburse the tax paid in VAT or will seek adjustments in the distribution of the tax burden among taxpayers. A critical point is the base of beneficiaries that will be used. This base must have credibility, as the program will be managed on it. In this sense, Barreix et al. (2012) recommend using conditional cash transfer programs that are already in use. Latin America has had a successful experience with these programs for decades, and, in general, they respect traditional criteria for defining the socially vulnerable population. To define its beneficiaries, the Devolve-ICMS Program uses the same registration as the Bolsa Família Program (PBF).

As for the individual amount to being refunded, several criteria can be applied. A first criterion is that it can be a fixed amount to be returned to the favored population. A second criterion would be regressive, which would consist in returning a percentage of the monthly purchases made electronically. This criterion favors the formal economy and improves tax equity and collection issues. This is considered a great virtue when there is a high level of informal employment. It needs significant investments in Information and Communication Technology (ICT). There is also the possibility of a hybrid model, with a fixed and variable portion. Tax refunds can be made on a monthly or quarterly basis by crediting to the target population's bank accounts or using an electronic debit card.



The use of account credit presents a positive side, which is the possibility of being able to withdraw money in cash; however, it can confuse with other transfers and not contribute to the proposal of raising awareness of tax citizenship materialized in the use of a specific card for the program.

Some studies have addressed the efficiency of the option to exempt items from the basic food basket as a way of mitigating the effects of regressive taxation. In 2018 and 2019, the Federal Government published two bulletins on the subject (Ministry of Economy 2019; Ministry of Finance 2018). To get an idea of the dimension of the tax waiver, the relief on the basic food basket is one of the highest tax expenditures of the Federal Government, and, in 2018, it presented an estimated cost of BRL 15.9 billion, equivalent to 5.4% of the total tax expenditure. It has been justified by its impact on the disposable income of the poorest. Reports estimate that the 10% poorest spend about 23.3% of their income on products exempt from the basic food basket, while the richest spend around 2.8%. It should be noted that the basic food basket exemption policy does not distinguish tax benefits by income level or type of food purchased, despite prioritizing low-income people who are more vulnerable to food insecurity and nutrition (Ministry of Economy 2019).

The basic food basket exemption adopts as a premise the hypothesis that the exemption would be passed on completely to the final prices of the contemplated products, which would allow cheaper products and increase access to the products in the basic food basket. However, it should be noted that this measure may not be effective in many situations, since the process of price formation in the economy depends on different market structures, the seasonality of agricultural production, and the price elasticity of food, among other factors. All this ends up influencing the focus. Some studies point to low elasticity for some products with a strong weight for the poorest, such as cereals, vegetables and tubers, pasta and bread, and chicken and eggs. In short, the exemption may not reach the intended final consumer (Ministry of Economy 2019).

The Ministry of Finance (2018) brought the historical context, economic fundamentals, fiscal dimension, international experiences, and a comparison of the results of this policy with income transfer programs. Based on a comparative analysis, the bulletin concludes that a direct income transfer focused only on the lowest-income population tends to be more efficient and effective for society than the basic food basket exemption policy, if the objective is to increase the welfare of the poorest.

The Ministry of Economy (2019) presented simulations of alternatives to the basic food basket exemption, comparing distributive effects. The tradeoff scenario highlights the total re-encumbrance of the basic food basket products and the total reallocation of its products to the PBF, and this demonstrated a more significant reduction in poverty and income inequality. An alternative proposal posed a scenario in which partial re-encumbrance of the basic food basket, with repercussions of 2.3% of the poorest 20% basket, and 11.2% of the richest 20% basket, would generate the amount of 1.2 billion, which, if applied to the PBF, would have the same effect in reducing poverty and income inequality. Even the scenario of egalitarian distribution of resources, without any focus, shows improvement in the indicators of poverty and inequality, which shows the regressive nature of the basic food basket exemption.

It is important to emphasize that, according to the Ministry of Economy (2019), in a scenario where the basic food basket exemption is only withdrawn without reallocating resources, there would be more people below the poverty line and an even greater effect on people below the extreme poverty line. It also points to a significant positive variation in the Gini index; that is, income inequality would also increase in this scenario. This is because the distribution of benefits from the basic food basket exemption, despite being regressive, is less concentrated than the distribution of the population's total income.

Araújo and Paes (2019) deepen the comparative analysis of the basic food basket versus the PBF, using a computable general equilibrium simulation. Their study used the neoclassical theoretical framework with discrete time, closed economy, and constant population and technology. In the model, there are two families differentiated by income

level: one that receives the PBF benefit and the other that does not, and the difference in income between these families is incorporated through the productivity parameter. The results showed that the increase in transfers generates a greater benefit to the well-being of the poorest class than the basic food basket exemption. The result persists for a targeting level of the PBF, as of 35.4%. For targeting levels lower than this, the basic food basket exemption is the best option. The 35.4% would be the balance point where the policies are equivalent. This emphasizes the importance of designing policies that maintain high targeting. Finally, it suggests the greater effectiveness of the PBF compared to the basic food basket exemption, regarding the objective of increasing the utility of the poorest.

## 4. Materials and Methods

To analyze the effectiveness of Devolve-ICMS, we used a difference-in-differences (DID) model. This approach is one of the most used when the objective is to measure the impacts of social policies or programs (Angrist and Pischke 2014; Ryan et al. 2015). The main advantage of this method over the others is that it allows the control of the differences between the analyzed units, including over time. This increases confidence that the results achieved are specifically due to the analyzed program. However, for this to occur, it is crucial to observe parallel trends in the response variable between the treated and control groups before the program's implementation. This is the main difficulty in properly using this method. Figure 3 shows that, in this study, this condition is met, which justifies its use. The *t*-test did not reject the hypothesis of equality of the average values of the response variables between the groups before the Program's implementation.

For this, two groups were constructed: treated and control, both for two periods, before and after the launch of the Program. The treated group is formed by eligible families that received the benefit, and the control group is formed by families with similar characteristics to those of the treated group, but that did not receive the benefit; in this case, they had the right to the card and did not seek it. The groups are restricted to Rio Grande do Sul, where the Program exists. The variables used are in Table 1.

**Table 1.** Variables used in the model.

| Acronyms | Description | Minimum | Average | Maximum | Source |
|:---:|:---|:---:|:---:|:---:|:---:|
| ref | Time in months (1, 2, 3, . . . , 26) | 1.00 | 11.36 | 26.00 | Sefaz/RS |
| CPF | Identifier of person/consumer | | | | Sefaz/RS |
| qtde_df | Monthly amount of Sefaz/RS tax documents | 0.00 | 1.06 | 109.00 | Sefaz/RS |
| vlr_dfr | Monthly real values of invoices per person (updated by IPCA) | 0.00 | 94.85 | 721.36 | Sefaz/RS |
| TNT | Dummy (treated = 1; untreated = 0) | 0.00 | 0.48 | 1.00 | Derivative |
| AD | Dummy (after = 1; before = 0) | 0.00 | 0.41 | 1.00 | Derivative |
| comvarv | Monthly index of retail sales volume in RS | 91.88 | 122.52 | 169.95 | IBGE |
| pimit | Monthly index of production volume of the manufacturing industry in RS | 95.44 | 109.27 | 120.13 | IBGE |

Source: Prepared by the authors.

The "ref" variable represents the months in which the data were consolidated, starting in November 2020, a year before the distribution of the Citizen Cards, and ending in December 2022, totaling 26 months. The "CPF" was used to identify the consumers. The variables "qtde_df" and "vlr_dfr" indicate the Program's response (outcome variables), with the first indicating the monthly number of tax documents issued to the CPFs of people in the treatment and control groups, and the second indicating the monthly amount corrected for December 2022 using the IPCA.

To define the groups, the month of November 2021, when the cards were delivered, was used as a cut-off point between before and after the Program (AD). The "TNT" indicates the treatment and control groups. The treatment group consisted of people who had the card in November 2021, whereas the control group consisted of people eligible for the

benefit and who did not seek their cards between November 2021 and December 2022. The last two variables, "comvarv" and "pimit", were used as control variables. The first indicates the retail sales volume and the second the industrial production volume, both from the state of Rio Grande do Sul. These two variables were used as indexes.

An average of 300,000 beneficiaries were analyzed monthly. On average, 48.5% are in the treatment group and 51.5% in the control group. In all, 7,795,039 records of tax documents were used. Values from documents considered outliers were removed. Before this adjustment, the database contained 8,983,955 records. This broad database enabled the use of the DID model, which is the most widely used method to assess the impacts of social programs. This method attributes to the intervention any difference in trends between the treatment and control groups that occur from the time the intervention begins. If other factors affect the difference in trends between the two groups, the estimate may be invalid or biased (Gertler et al. 2015).

The existence of a large group of eligible people who did not seek the card allowed the formation of a control group with characteristics similar to those of the treatment group (eligible people who sought the card). Without the card, the beneficiary does not receive the tax refund. The difference is that one group started benefiting from the Program and the other did not. This is important because it guarantees that the characteristics of the groups are similar, increasing the reliability of the model results. The use of control variables also supports the robustness of the results.

Possible reasons for eligible people not picking up the card: short time of Program implementation, and little information about the Program in the media. There is no reason to believe that this fact represents differences between the treated and control groups in such a way as to bias the results. In addition, both people in the control group and those in the treated group are part of the Single Registry—a register of people living in poverty and extreme poverty used by the Federal Government to implement social programs.

The validity of the underlying assumption of the equality of trends can be assessed, although it cannot be proved. A good test of the validity of this hypothesis is to compare changes in response variables for the treatment and control groups before the Program's implementation. If the outcome variables moved together for both groups, greater confidence is ensured that they would follow the same trend in the post-intervention period (Gertler et al. 2015). Figure 3 shows that the behavior of the two groups is similar before the Program. This increases confidence that the differences observed after the Program's implementation are due exclusively to its effect (Angrist and Pischke 2009; Rosenbaum and Rubin 1983).

The estimated model consisted of an unbalanced panel, because not all CPFs had records every month. Three panels were run: stacked, random effects, and fixed effects. In all cases, the Chow, Breusch-Pagan, and Hausman tests indicated that Fixed Effects modeling was the best, controlling for unobservable factors that are invariant over time. Furthermore, the models were estimated in a robust form to correct heteroscedasticity problems. In formal terms, the DD models used in this study can be written as follows:

$$\text{vlr\_dfr} = \alpha + \beta_1 \times AD + \beta_2 \times TNT + \beta_3 \times DID + \beta_4 \times \text{comvarv} + \beta_5 \times \text{pimit} + \varepsilon \quad (1)$$

$$\text{qtde\_df} = \alpha + \beta_1 \times AD + \beta_2 \times TNT + \beta_3 \times DID + \beta_4 \times \text{comvarv} + \beta_5 \times \text{pimit} + \varepsilon \quad (2)$$

where: $\alpha$, $\beta_1$, $\beta_2$, $\beta_3$, $\beta_4$, and $\beta_5$ are the estimated parameters, DID represents the AD$^x$TNT interaction, and $\varepsilon$ is the error term. The other variables are defined in Table 2. Variable subscripts have been omitted. The parameter of interest in the model is $\beta_3$, and the hypothesis is that it is positive and significant in both cases. For the first case, a positive $\beta_3$ means an increase in the monthly consumption of the treatment group concerning the control group. In the second case, a positive $\beta_3$ means an increase in the monthly number of tax documents.

**Table 2.** Summary of the amount returned by the Program by minimum wage ranges.

| Information | <1 M.W. | 1–2 M.W. | >2 M.W. | Total (Mean) |
|---|---|---|---|---|
| Number of beneficiaries | 412,572 | 17,970 | 3377 | 433,919 |
| Families in the income range (%) | 95.08% | 4.14% | 0.78% | 100% |
| Average monthly income (R$) | 217.47 | 1634.24 | 2895.16 | 296.98 |
| The average monthly amount returned (R$) | 35.93 | 71.72 | 85.93 | 37.81 |
| Amount Returned/Income (%) | 16.5% | 4.4% | 3.0% | 12.7% |

Source: Prepared by the authors based on data from SEFAZ/RS (Revenue Office of the state of of Rio Grande do Sul).

## 5. Results and Discussion

Tax refunds, as well as operational and costing expenses, are paid through the state's budget allocations, currently around BRL 200 million per year. As for the beneficiaries' profile, the data show that most of them are in the 31–40 age range (173,400 people), followed by the 21–30 age range (155,000 people). In terms of gender, the vast majority are women, accounting for 82.8% of all beneficiaries. As for the expenditure profile, the data indicate that 83% of the resources were used in the purchase of necessities in supermarkets, wholesales, butcheries, restaurants, and bakeries. Another 5.9% went to health products and services. Regarding the importance of the amounts returned, the data in Table 2 show that, for people with an income of up to one minimum wage (95% of the total), refunds represented more than 16% of their income.

Brazil has a complex fiscal and tax system with extremely relevant indirect taxation on consumption regarding collection. One of the effects of indirect taxation is fiscal anesthesia, which distorts the taxpayers from the perception of the tax they bear. Another important characteristic of consumption taxation is its regressivity, which consists of a greater tax burden on those with lower incomes. Therefore, the decrease in tax regression contributes to improving the living conditions of the poorest citizens. With the database of this study, it was possible to determine the tax pressure profile (ICMS/Income) after the personalized returns of the Devolve-ICMS Program.

Figure 2 shows that the observed effects are fully in line with the expectations and objectives of the Program; that is, it has established fiscal progressivity in an originally regressive system. It can be noticed that the incidence, previously regressive (blue line), starts to behave in a markedly progressive way (orange line), especially in the range of up to two minimum wages, and slightly progressive (close to neutrality) from this range onwards[2].

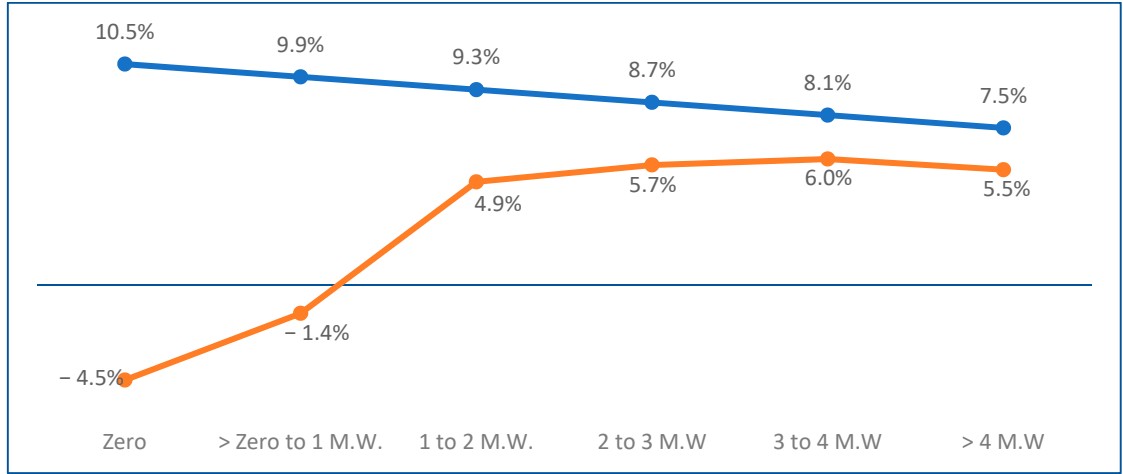

**Figure 2.** Fiscal pressure by minimum wage brackets before and after the Devolve-ICMS Program. Note: The top blue line represents the time "before" the ICMS refunds; the bottom orange line corresponds to the time "after" the ICMS returns. Source: SEFAZ/RS.

Figure 3a shows the similarity of behavior between groups regarding the value of tax documents before the start of the Program in November 2021 (month 13 on the horizontal axis). As of this month, the value varies between BRL 19.00 and 48.00 per month more in the treatment group. Figure 3b shows the behavior of the groups regarding the number of tax documents issued. Again, there is a great similarity in behavior before the Program and important differences after its implementation. From November 2022 (month 13), a higher monthly number of documents can be observed in the treatment group, ranging from 21 to 46%.

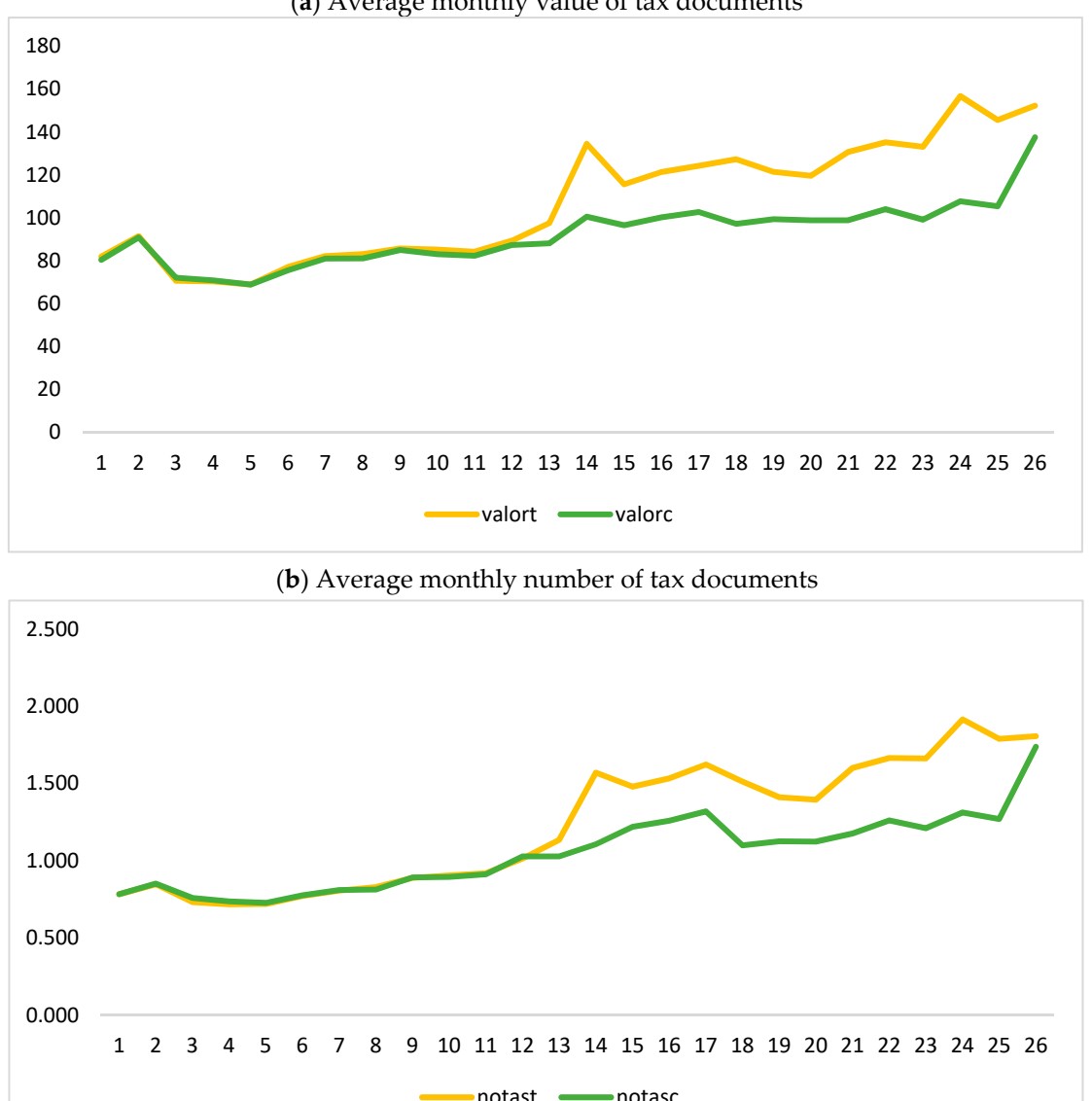

**Figure 3.** The behavior of the treatment and control groups before and after the implementation of the Devolve-ICMS Program. Notes: valort = monthly value for the treatment group; valorc = monthly value for the control group; notast = monthly number of tax records issued by CPF in the treatment group; and notasc = monthly number of tax records issued by CPF in the control group. Source: Prepared by the authors.

Figure 3 provides evidence that the Program had positive impacts on the treatment group, in terms of the monthly amount and the monthly number of tax documents. The DID model enables statistical verification and quantification of these impacts. Regarding the impacts estimated using this method, Table 3(1) shows the results for the monthly value

of invoices. The model estimated that the treatment group had a monthly consumption of over BRL 32.47. This means that the Devolve-ICMS Program's resources have increased their living conditions, since the resources were not used to pay debts or to any third parties, but resulted in an increase in their consumption. This was the expected result and matches the high focus of the Program.

**Table 3.** Results of regressions using fixed effects panel (robust).

|  | Value of Invoices (vlr_dfr) | Quantities of Invoices (qtde_df) |
|---|---|---|
|  | **(1)** | **(2)** |
| AD | 18.8270 *** | 0.3666 *** |
|  | (0.2205) | (0.0032) |
| DID (ATET) | 32.4728 *** | 0.4390 *** |
|  | (0.3228) | (0.0048) |
| comvarv | 0.7220 *** | 0.0050 *** |
|  | (0.0047) | (0.0000) |
| pimit | 0.1496 *** | 0.0028 *** |
|  | (0.0087) | (0.0001) |
| Observations | 7,795,039 | 7,795,039 |
| *R*-squared | 0.0279 | 0.0435 |
| *F*-statistic | 52,840 *** | 83,668 *** |

Notes: Standard errors are in parentheses. * $p < 0.10$, ** $p < 0.05$, *** $p < 0.01$. The coefficient associated with the DID variable is the ATET (average treatment effect on the treated). Source: Prepared by the authors.

Regarding the number of documents issued with a CPF, verified in the invoice database, the model estimated that the treatment group had a monthly issuance of 0.43 documents (Table 3(2)). This means that the Program encouraged the expansion of the beneficiaries' participation in the inclusion of their CPF in the invoice, indicating an improvement in fiscal education and, consequently, greater fiscal citizenship.

One of the pillars of tax education is the taxpayers' engagement in requesting companies to issue invoices. Invoice issuance ensures the registration of taxes collected from the consumer, and the government can control the payment of taxes by companies. Without this engagement, the government collects less, which may lead to restrictions on the supply of a series of public services, such as education, health, and security. With a low-tax education, everyone is penalized.

## 6. Conclusions

The first objective of this study was to evaluate whether the Devolve-ICMS Program managed to reduce the tax pressure on the low-income population. Using data from the consumption tax and income of the population, it was verified that the Program has reached this objective. It relatively lessens the fiscal pressure on the population in lower income brackets, establishing progressivity in the tax incidence.

The second objective was to estimate the impacts of the Program on the value of consumption and the number of tax documents issued to beneficiaries. For this, the difference-in-differences (DID) method was used. To this end, two models were estimated, and the results showed that the beneficiaries had a monthly increase in consumption expenses and the number of invoices issued. These results suggest that the Program has improved the living conditions of the neediest population in the state of Rio Grande do Sul.

Other consequences of this Program were an increase in the formalization of economic transactions and an improvement in fiscal education. This greater engagement in requesting invoices increases government revenue, which may result in expansion and improvement in the provision of public services.

Other social programs seek to improve the economic conditions of the neediest citizens, such as the *Bolsa Família* Program and the basic food basket exemption. This study brings evidence of an innovative program on the national scene, the Devolve-ICMS, whose

objectives are similar to those of previous programs, but with different characteristics. This is a focused program—applied at a subnational level—that consists in refunding the tax paid on consumption by the poorest people. The results obtained indicate that this Program has the potential to be expanded to other states of the Federation and, thus, be applied at the national level, as well as in other countries. The greatest virtue of the Devolve-ICMS Program is in its focus, resulting in efficiency gains in the use of public resources.

A limitation of the study is the relatively short period of analysis, since the Program was implemented approximately 12 months ago. Therefore, the trend is that, in the coming months, there will be an increase in the number of beneficiaries, greater engagement in requesting invoices, and higher consumption value. The idea is to continue evaluating the impacts of this Program, including new periods in the analysis, as well as other response variables, such as information on nutritional quality and inequality. In any case, the database used in this study is quite expressive, providing safety regarding the results.

**Author Contributions:** Conceptualization, J.L.T.; methodology, J.L.T., A.F. and G.P.d.S.; investigation, J.L.T., A.F. and G.P.d.S.; writing—original draft preparation, J.L.T.; writing—review and editing, J.L.T., A.F. and G.P.d.S. All authors have read and agreed to the published version of the manuscript.

**Funding:** This research received no external funding.

**Data Availability Statement:** Data is protected by tax secrecy.

**Conflicts of Interest:** The authors declare no conflict of interest.

## Notes

[1]    The *Cadastro Único* is a set of data on Brazilian families living in poverty and extreme poverty.
[2]    To determine the initial fiscal pressure, that is, before refunds, it is necessary to estimate the tax borne by the beneficiary families by income range, which is obtained by applying the rates on consumption estimated in the Family Budget Survey (POF/IBGE). Therefore, to determine the final fiscal pressure, the individual amount of refunds is deducted from the initial pressure.

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
