# Peer review of "Refund of Consumption Tax to Low-Income People: Impact Assessment Using Difference-in-Differences"

_economies, doi:10.3390/economies11060153_

Round 1

Reviewer 1 Report

This study estimates the effect of a tax refund policy in Brazil and finds that the tax refund to lower-income people significantly increases targeted people's consumption. The research question is important but I have several concerns regarding the paper, listed below.

Regarding the identification strategy, the study compares people eligible for tax refund, but having or not having the card. My question is, if the card is important and useful in many cases, why people do not apply for the card? If the people with and without the card are different (for example, the latter may be less aware of benefitting policies), the identification design is weakened.

The author answer the first-stage question but what's next in temrs of economic stories? For example, while the tax refund policy increases the targeted people's consumption, how does the welfare or the inequality level change correspondingly?

The introduction is less informative. The author is suggested to start by emphasizing a general question (i.e., why researching on tax refund to lower income people is important). Sentences like "Two groups were constructed: treatment and control." can be dropped, as the readers are professional and shall be familiar with the textbook-level DID design. 

There are too many fonts in tables and equations. The author should keep the content uniformally written.

Tables and graphs are less formal in terms of format. I urge the author to follow the styles of some leading economic journals, such as AER or QJE.

The language is not easy to read. Some expression might be misleading. A proofreading by native speaker is suggested.

Author Response

Reviewer 1

Regarding the identification strategy, the study compares people eligible for tax refund, but having or not having the card. My question is, if the card is important and useful in many cases, why people do not apply for the card? If the people with and without the card are different (for example, the latter may be less aware of benefitting policies), the identification design is weakened.

Answer: We thank the reviewer for this comment. This paragraph has been included in the manuscript to clarify this point. “Possible reasons for eligible people not picking up the card: short time of program implementation and little information about the program in the media. There is no reason to believe that this fact represents differences between the treated and control groups in such a way as to bias the results. In addition, both people in the control group and those in the treated group are part of the Single Registry – a register of people living in poverty and extreme poverty used by the federal government to implement social programs”.

The author answers the first-stage question but what's next in terms of economic stories? For example, while the tax refund policy increases the targeted people's consumption, how does the welfare or the inequality level change correspondingly?

Answer: The analyzed program aims to combat inequality and improve the living conditions of the poorest population. In our study, the objective was to measure the impact of the program on the value of consumption and the number of tax documents. Thus, welfare and inequality enter the analysis indirectly. We have tried to make this clearer in the text.

The introduction is less informative. The author is suggested to start by emphasizing a general question (i.e., why researching on tax refund to lower income people is important). Sentences like "Two groups were constructed: treatment and control." can be dropped, as the readers are professional and shall be familiar with the textbook-level DID design.

Answer: We thank the reviewer for this comment. The introduction has been revised. Some sentences, such as the one indicated, were reformulated or withdrawn.

There are too many fonts in tables and equations. The author should keep the content uniformly written.

Answer: The text has been standardized.

Tables and graphs are less formal in terms of format. I urge the author to follow the styles of some leading economic journals, such as AER or QJE.

Answer: The style of graphs and tables has been adapted to the QJE standard.

The language is not easy to read. Some expression might be misleading. A proofreading by native speaker is suggested.

Answer: The text was edited for proper English language, grammar, punctuation, spelling, and overall style by an accredited editor.

Reviewer 2 Report

Review of the paper titled „Refund of consumption tax to low-income people: impact assessment using differences in differences” under consideration to the Economies Journal:

Major issues:

- In the introductory section, the novelty of the paper could be described in a more straightforward way.

- Please add a short summary of the main results of the study before the agenda of the paper in the introductory part.

- Materials and methods: here you should briefly describe the advantages and disadvantages of using diff-in-diff, especially when it comes to endogeneity, omitted variable bias, sample selection, etc. The method have some drawbacks, which could be described in the paper.

- One should be more precise about the conditions on which the control group was formulated. Similarly, have the initial differences between the treatment and control groups been tested? Etc. t-test.

- The most important formal test (parallel pre-treatment trends) seems to be validated, which is good.

- In Table 1, you could add some brief information about the descriptive statistics.

- The author(s) could refer to terms such as ATE (average treatment effect) or ATET (average treatment effect on the treated), e.g. in Table 3.

- You should work on the discussion of the results, perhaps with similar income-related programmes.

Minor issues:

- Consider reviewing the type of references in the manuscript to see if they meet the journal's requirements.

Final thoughts:

The paper is well organised and presents interesting findings. The quality of the research is good, with only minor points that need to be corrected. The authors should concentrate their efforts on discussing their findings with other/similar evaluations of the programmes, if possible, together with implementing some improvements to the methodological part. Therefore, I suggest a minor revision.

Author Response

In the introductory section, the novelty of the paper could be described in a more straightforward way.

Answer: We thank the reviewer for this comment. The introduction has been revised in order to make the contribution of the article clearer.

Please add a short summary of the main results of the study before the agenda of the paper in the introductory part.

Answer: A paragraph with the main results was included before the last paragraph of the Introduction section.

Materials and methods: here you should briefly describe the advantages and disadvantages of using diff-in-diff, especially when it comes to endogeneity, omitted variable bias, sample selection, etc. The method has some drawbacks, which could be described in the paper.

Answer: We thank the reviewer for this comment. An excerpt pointing out the advantages and conditions for using the method has been included in the Materials and Methods section.

One should be more precise about the conditions on which the control group was formulated. Similarly, have the initial differences between the treatment and control groups been tested? Etc. t-test.

Answer: We have improved the explanation on this matter and performed the t-test.

In Table 1, you could add some brief information about the descriptive statistics.

 Answer: This information has been included.

The author(s) could refer to terms such as ATE (average treatment effect) or ATET (average treatment effect on the treated), e.g., in Table 3.

Answer: The information has been included.

You should work on the discussion of the results, perhaps with similar income-related programmes.

Answer: We have not found studies that evaluated similar programs applied in Japan and Canada.

Consider reviewing the type of references in the manuscript to see if they meet the journal's requirements.

Answer: The references has been reformatted.

Round 2

Reviewer 1 Report

I have no more questions. I think that the author answered most questions that I and the other referee had raised.